# Recent Advances in Regioselective C–H Bond Functionalization of Free Phenols

**DOI:** 10.3390/molecules28083397

**Published:** 2023-04-12

**Authors:** Yanan Li, Yekai Huang, Zhi Li, Jianan Sun

**Affiliations:** 1School of Chemical Engineering, Anhui University of Science and Technology, Huainan 232001, China; 2School of Biomedical Engineering, Anhui Medical University, Hefei 230032, China

**Keywords:** free phenol, C–H functionalization, regioselectivity, synthesis, cross-coupling

## Abstract

Phenols are important readily available synthetic building blocks and starting materials for organic synthetic transformations, which are widely found in agrochemicals, pharmaceuticals, and functional materials. The C–H functionalization of free phenols has proven to be an extremely useful tool in organic synthesis, which provides efficient increases in phenol molecular complexity. Therefore, approaches to functionalizing existing C–H bonds of free phenols have continuously attracted the attention of organic chemists. In this review, we summarize the current knowledge and recent advances in *ortho*-, *meta*-, and *para*-selective C–H functionalization of free phenols in the last five years.

## 1. Introduction

Phenols are readily available and have emerged as ubiquitous building blocks in natural products, pharmaceuticals, and functional materials [1,2,3,4,5,6]. For example, many natural products such as hormones, antibiotics, vitamins, and neurotransmitters are derived from or contain phenols; Approximately, more than 10% of the top 200 selling pharmaceuticals bear at least one phenol or employ phenols as synthetic intermediates; phenols are also key components of functional materials such as polymer resins (Figure 1). Consequently, straightforward and highly efficient site-selective C–H bond functionalization of free phenols has continuously attracted the attention of organic chemists [7,8,9,10].

With developments made in organic synthesis, the direct C–H functionalization of phenols has been explored extensively as an efficient and powerful tool to synthesize complex phenol molecules. In the previous selective conversion of phenol, most of the directing groups installed on the phenolic hydroxyl group were used to realize the indirect strategy of activation of *ortho* C–H bond functionalization. However, as the most straightforward and atom-economical strategy, directly using free phenolic hydroxyl groups as directing groups to synthesize highly substituted phenolic derivatives, there are many challenges for chemo- and regioselective C–H functionalization of unprotected free phenols. The major barrier stems from several aspects: (1) the phenolic hydroxyl group has strong acidity and nucleophilicity, so that the reaction site mostly occurs on the phenolic hydroxyl group instead of the benzene ring; (2) the *ortho* and *para* positions of the phenolic hydroxyl group in the benzene ring are highly active, which reduces the regioselectivity of the C–H bond functionalization reaction; (3) it is often necessary to add a stoichiometric oxidant in the C–H functionalization reaction, and the phenols are typical electron-rich aromatic hydrocarbons, which are easily oxidatively decomposed during the reaction. Thus, more economical and green synthetic approaches for the direct regioselective C–H bond functionalization of free phenols are a significant challenge of great interest.

Some reviews on the transition-metal-catalyzed functionalization of C–H bonds in phenols have been reported so far. Lumb and co-workers [11] wrote a review on phenol-directed C–H functionalization in 2018. Luo [12] summarized the progress of the transition metal-catalyzed directing-group-assisted C–H activation of phenols until 2019. Immediately afterward, Mamari [13] outlined a comprehensive advance in regard to metal-catalyzed C–H bond functionalization of phenol derivatives in 2020. Later, Youn and co-workers [14] reported a comprehensive overview of transition-metal-catalyzed *ortho*-selective C–H functionalization reactions of free phenols until 2021. Recently, Zhai [15] published an excellent review on recent advances in catalytic oxidative reactions of phenols and naphthalenols. We want to focus on the recent development on the C–H functionalization of different positions on the free phenol aromatic ring. Thus, we will summarize recent advances since 2018 in regioselective C–H bond functionalization of free phenols in this review. This review is classified into three sections according to the position of functionalization, treating *ortho*-, *meta*-, and *para*-positions of free phenol successively, and only some selected examples are described schematically (Figure 1).

## 2. C2-Functionalization

### 2.1. Metal-Catalyzed C–H Bond Functionalization of Free Phenol

Over the past decades, there have been tremendous advances in the field of catalytic C–H bond functionalization [16,17,18], among which metal-catalyzed C–H bond functionalization of free phenol has proven to be an extremely useful tool in the synthesis of complex phenol molecules. 

#### 2.1.1. Cu-Catalyzed C–H Bond Functionalization 

In 2018, Jain’s group [19] synthesized a GO-Cu_7_S_4_NPs from Cu_2_S to achieve an highly efficient *ortho-*selective C–H aminomethylation of free phenol derivatives **1** with *N,N*-dimethylbenzylamines **2** using *tert*-Butyl hydroperoxide (TBHP) as an oxidant under solvent-free conditions. The nano-copper catalyst could be reused for C_sp2_-C_sp3_ cross-dehydrogenative coupling, with no requirement for pre-functionalization of substrates. A variety of C–C coupled products **3** are obtained with 68–89% yields. Furthermore, more challenging substrates like β-naphthols and substituted halohydroxypyridine derivatives were successfully used in the aminomethylation reaction to afford the corresponding aminomethylated products in 70–80% yields. Control experiments suggested that this reaction may undergo the nucleophilic attack by *ortho*-phenol carbon **4** on iminium ion **5**, which was formed by the loss of hydrogen atom via a single electron transfer under the action of the nano-copper catalyst and TBHP, to generate the Cu-coordinated ketone **7** (Figure 2).

Simultaneously, Patureau’s group [20] reported a highly selective Cu(II)-catalyzed cross-dehydrogenative *ortho*-aminomethylation of free phenols **8** with aniline derivatives **9** using Di-*tert*-butyl peroxide (DTBP) as an oxidant. As illustrated in Figure 3, a series of aminomethylated products **10** were obtained with 32–82% yields and based on controlled experiments; the authors proposed that this reaction involved the aminomethyl radical **13** by a Cu-centered six membered transition state **15**. However, this method required an excess of amines (6.3–9.5 equiv.) and only tertiary *N*-methyl aromatic amines were compatible with the reaction system, which limited its synthetic utility.

In 2021, a copper(II)-modified nitrogen-rich covalent organic polymer (Cu/N-COP-1) was prepared and employed to achieve the cross-dehydrogenative *ortho*-aminomethylation of free phenols **16** with *N,N*-dimethylanilines **17** under mild conditions by Xie et al. [21]. Various aminomethylated products **18** were obtained in 42–59% yields (Figure 4). In addition, the catalyst could be easily reused for at least five consecutive runs without significant loss of catalytic activity, which is attributed to the interaction between the nitrogen-containing group of the covalent organic polymer and copper chloride. Having the di-substituted products **19** and moderate yields (mostly 42–59%) of the desired products were the drawbacks of this method.

In 2020, Ma et al. [22] reported the first copper-catalyzed highly efficient *ortho*-C–H bond functionalization rather than O–H insertion of free phenols **20** with *a*-aryl-*a*-diazoesters **21** (Figure 5). A range of alkylation products **22** were obtained in 76–98% yields under mild reaction conditions. A preliminary mechanistic study indicated that the hydroxyl group was important not only for site-selectivity but also for the reactivity of the C–H bond functionalization reaction, and the *ortho*-selectivity could be improved via the interaction between the hydroxyl group and the copper catalyst. Herein, the authors proposed two possible reaction mechanisms for this transformation, which involved the Cu-carbene **27** and the carbocation intermediate **28** formed by coordination between the CuCl_2_ and the nitrogen of *a*-aryl-*a*-diazoesters, respectively.

One year later, Xie’s group [23] developed a copper-catalyzed tandem cross-coupling/annulation of free phenols **30** with ketoximes **29** via dual C–H functionalization. A variety of oximes, including 1-naphthyl-bearing oxime and heteroaryl-bearing oximes, were compatible for this reaction system and afforded the corresponding products **31** in good yields. Control experiments disclosed that the benzoquinone **33** might be the vital intermediate of the dual C–H functionalization reaction, as the *para-*dihydroxyl was crucial in the process of reaction (Figure 6).

In 2022, Zhao et al. [24] reported a copper-catalyzed C2-site selective amination of free *p*-aminophenol derivatives **35** with arylamines **36** under mild conditions, using air as a terminal oxidant, and a variety of C2-site functional aminophenol derivatives **37** were obtained with 39–78% yields (Figure 7). Interestingly, the *para*-benzomorpholine substituted phenol was also compatible with the reaction system for transformation into the desired C2-site selective aminated product **37a** in 58% yield with oxidation of the benzomorpholine group to *4H*-benzo[*b*][1,4]oxazine. Based on controlled experiments, the authors proposed that the amination reaction may undergo a process of radical-radical cross-coupling between the phenoxyl radical **39** and the *N*-radical **40**.

#### 2.1.2. Co-Catalyzed C–H Bond Functionalization 

In 2018, Khakyzadeh, Zolfigol, and co-workers [25] disclosed a Co(II)-catalyzed regioselective synthesis of 2-(aryl/alkylthio)phenols **44** via formal C–H bond functionalization of free phenols **42** with aryl/alkyl thiols **43** using acetic anhydride as the directing group under mild conditions. A variety of 2-(aryl/alkylthio)phenols **44** were obtained with moderate to excellent yields (Figure 8). In addition, 2-naphthol and 1-naphthol were also compatible with this reaction system. Later, this reaction was further developed through Fe_3_O_4_@SiO_2_-UT@Co(II), using pivalic anhydride as the directing group, by Khaef et al. [26]. In fact, the two reactions were stepwise processes. By introducing directing groups to protect the hydroxyl group, the *ortho*-C–H functionalization of phenol was achieved. Finally, 2-(aryl/alkylthio)phenols were obtained by an acidic hydrolysis strategy. Depending on the control experiment, both thiolation reactions may undergo a process of free radicals **45**. 

In 2019, a Cobalt(II)[salen]-catalyzed selective aerobic oxidative cross-coupling of electron-rich free phenols **46** with 2-naphthols **47** was developed by the Pappo group [27] in a recyclable 1,1,1,3,3,3-hexafluoropropan-2-ol (HFIP) solvent. A variety of nonsymmetric biphenols **48** were obtained with 40–92% yields under mild waste-free conditions by a selectivity-driven catalyst design approach. However, the nonsymmetric biphenols **50**, **51** and **52** were obtained by the *para*-C–H bond functionalization rather than *ortho*-positions of free phenols with 2-naphthols (Figure 9). Besides, a strong electron-withdrawing group such as ester was not compatible with the current system, failing to afford the corresponding biaryl coupling product. Detailed control experiments and kinetic studies suggested that a liberated phenoxyl radical **53** and a ligated 2-naphthoxyl radical **54** were involved in the reaction pathway.

### 2.2. Electrochemical/Photochemical C–H Bond Functionalization 

Electrochemistry/photochemistry has been a powerful synthetic tool for organic chemistry [28,29,30,31,32,33]. Compared with the chemical methods of the C–H functionalization of free phenols, the electrochemistry/photochemistry method uses electrons/light as a renewable and environmentally friendly reagent to replace the metal catalysts and the stoichiometric amounts of oxidant in the C–H functionalization reactions. Undoubtedly, the electrochemical/photochemical C–H bond functionalization of free phenols has attracted increasing attention from organic chemists.

The highly efficient arylation of benzothiophenes **55** with free phenols **56** was realized by the Waldvogel group [34] under exogenous oxidant-free and metal catalyst-free electrochemical oxidation conditions in 2018. The 2- or 3-(hydroxyphenyl)benzo[*b*]thiophenes **57** were regioselective, obtained with 38–88% yields in an undivided cell equipped with a BDD anode and a BDD cathode at a constant current of 5.2 mA/cm^2^, using Bu_3_NMeO_3_SOMe as the electrolyte at RT or 40 °C (Figure 10). Later, Yue et al. [35] developed an electrochemical oxidative C–H/C–H coupling of free phenols **58** with 3-phenylbenzothiophenes **59**, which delivered highly tunable benzothiophene derivatives **60** with 33–89% yields in an undivided cell equipped with a RVC anode and a Pt plate cathode at a constant current of 10 mA, using *n*-Bu_4_NPF_6_ as the electrolyte at 25 °C under the protection of nitrogen (Figure 10). Both monofunctional and bifunctional groups could be achieved in this electrochemical oxidation reaction under external oxidant- and catalyst-free conditions. However, the thiophene and benzothiophene were not compatible with the reaction system. Control experiments suggested that a cross-coupling process between the *p*-methoxylphenol radical **62** and the 3-phenylbenzothiophene radical cation **63** might be involved in this electrochemical transformation.

In 2018, the Lei group [36] reported an electrochemical oxidative C–H amination of free phenols **65** with phenothiazine derivatives **66** under metal catalyst- and chemical oxidant-free conditions. A host of *N*-aryl phenothiazines **67** were obtained with 17–93% yields in an undivided cell equipped with a graphite rod anode and a nickel plate cathode at a constant current of 7 mA, using *n*-Bu_4_NBF_4_ as the electrolyte at room temperature under nitrogen protection (Figure 11). Notably, 2-naphthol and 1-naphthol were also compatible with this reaction system, the products of the *ortho*-C–H bond functionalization of 2-naphthol and the *para*-C–H bond functionalization of 1-naphthol were obtained in 89% yield and 88% yield, respectively. Besides, the electro-oxidative double C–H amination of *ortho*-substituted phenols was also observed in this reaction. Based on cyclic voltammetry (CV) experiments, the authors proposed that the electrocatalytic amination reaction involved the oxidation of amine substrate to generate the radical cation **70**.

Subsequently, Feng et al. [37] reported an electro-oxidative and regioselective C–H azolation of free phenols **72** under external oxidant-free conditions with H_2_ evolution. The electrosynthesis proceeds in a simple undivided cell equipped with two platinum electrodes at a constant potential of 2.5 V, using *n*-Bu_4_NPF_6_ as an electrolyte at room temperature under the protection of nitrogen. Moreover, the *para*-methoxyl aniline derivatives were successfully employed as the reaction substrate under standard conditions. Detailed experiments and cyclic voltammetry studies suggested that a radical coupling process between the carbon radical **76** and the nitrogen-centered radical **77** might be involved in this electrochemical transformation (Figure 12).

In 2019, Sun, Zeng, and co-workers [38] reported a selective electro-oxidative cross-coupling of different free phenols **79** and naphthols **80** using tri(*p*-bromophenyl)amine(TBPA) **82** as a redox mediator. A variety of non-symmetric biphenols **81** were obtained with 27–83% yields in an H-type divided cell equipped with a Pt plate anode and a graphite plate cathode at 0.8 V vs. Ag/AgNO_3_, using LiClO_4_ as the electrolyte (Figure 13). When 2,6-dimethoxyphenol **84** was used as a substrate, the *para*-C–H bond functionalization of 2,6-dimethoxyphenol was observed. Based on the CV analyses, the authors proposed that this cross-coupling reaction started from an electrochemical oxidation of TBPA **82** at the surface of anode to generate its cation radical TBPA^+•^ **83**. Then, homogeneous electron transfer of TBPA^+•^ with 2,6-dimethoxyphenol **84** formed the cation radical **85**, which underwent nucleophilic addition with 2-naphthol.

Afterwards, the Waldvogel group [39] further developed an electrochemical dehydrogenative coupling reaction of free phenols **86** carrying electron-withdrawing groups using DIPEA as an additive. The homo-coupling reaction was conducted in an undivided cell equipped with BDD electrodes at a constant current of 5 mA/cm^2^, lacking a supporting electrolyte at room temperature. Besides, this method could be extended to cross-coupling reactions with naphthalenes to form biaryls and precursors for dibenzofurans. Nevertheless, the moderate yield of the products was the drawback of this method. Control experiments suggested that the C–C coupling reaction may involve the coupling of free radicals (Figure 14).

In 2021, Wen, Yang, and co-workers [40] reported an electrochemical in situ oxidative sulfonylation of free phenols **88** with sulfinic acids **89** to give sulfonylated hydroquinones **90** with 32–95% yields in an undivided cell equipped with a graphite rod anode and a platinum plate cathode at a constant current of 10 mA, using LiClO_4_ as the electrolyte (Figure 15). When the applicable scope of phenols with different substituents was explored under standard conditions, the regional selectivity of the sulfonylated reaction is slightly poor, due to the steric and electronic effect of the phenols. Besides, the aliphatic sulfinic acids were not compatible with this reaction system. Based on controlled experiments, two postulated reaction pathways were involved in sulfonylated reactions, which were radical-radical cross-coupling between the sulfone radical **98** and the carbon radical **95** or the anodic oxidation of hydroquinone to form benzoquinone **92** followed by Michael addition with sulfinic acids.

In 2021, Li and co-workers [41] reported a catalyst-free visible-light-driven *ortho*-C(sp^2^)–H arylation of free phenols **100** with arylbromides **99** under irradiation of a blue LED or natural sunlight. A series of 2-arylated phenols **101**, including the electron-donating and electron-withdrawing groups, were obtained with 38–73% yields at room temperature (Figure 16). However, anisole was not successfully used in the arylation reaction to afford the corresponding product, and heteroaryl halides were also incompatible with this reaction system. Based on controlled experiments, the authors proposed that the reaction proceeded via visible light photoexcitation of an EDA complex **102** between an aryl bromide and a phenolate ion.

Subsequently, a photocatalytic oxidative coupling of free phenols **103** with alkenylphenols was reported, using a recyclable heterogeneous titanium dioxide photocatalyst in air and visible light, by Kozlowski group [42]. When 2,6-dimethylphenol and 2-isopropyl-5-methylphenol were used as substrates in homo-coupling reaction, the *para*-C–H bond functionalization was observed. The incomplete conversion with moderate yields of the desired products limited the application of the reaction. The bare leak of the phenol hydroxyl group played a key role in the reaction, since anisole was incompatible with this reaction system. Depending on the control experiment, the authors proposed that this reaction may involve the TiO_2_–phenol complex **105**, which was activated through a ligand to metal charge transfer effect (LMCT) (Figure 17).

### 2.3. Others

In 2018, Li and co-workers [43] reported a B(C_6_F_5_)_3_-catalyzed hydroarylation of free phenols **106** with 1,3-dienes **107** under mild reaction conditions. A series of structurally diverse *ortho*-allyl phenols **108** were obtained with 32–91% yields. When the phenol was applied as the substrate, a small quantity of *para*-allylation products was obtained. Moreover, 1,4-dimethoxybenzene was not successfully used in the hydroarylation reaction to afford the corresponding product. A preliminary mechanistic study suggested that the hydroarylation reaction took place via a borane-promoted protonation/Friedel-Crafts pathway, which involved a π-complex of carbocation-anion contact ionpairs **109** (Figure 18). Subsequently, this method was developed by Zhou et al. to realize a B(C_6_F_5_)_3_ catalyzed hydroarylation of free phenols **110** with terminal alkynes **111** at room temperature (Figure 18) [44]. As compared with hydroarylation of 1,3-dienes, the difference was that when 2,6-disubstituted free phenol was employed as the substrate, no *para*-hydroarylation product was observed in the hydroarylation of terminal alkynes with free phenols.

In 2020, Lei’s group [45] reported a K_2_S_2_O_8_ -induced azolation of electron-rich free phenol derivatives **114** with pyrazoles **115** under catalyst-free conditions. A variety of *N*-arylazoles **116** bearing the electrondonating groups were obtained with 20–76% yields via oxidant-induced strategy. Detailed experiment studies suggested that a radical coupling process between the *N*-centered radical **117** and the C-centered radical **118** might be involved in this azolation transformation, which was generated by the oxidation of K_2_S_2_O_8_ (Figure 19).

In 2021, Liu and co-workers [46] reported a Brønsted acid catalyzed chemo- and *ortho*-selective aminomethylation of free phenols **120** with *N,O*-acetals **121** under mild conditions. Various aminomethylated phenol products **122** were obtained with 42–97% yields. The high *ortho*-selectivity was attributed to the formation of an intermediate **123** via the interaction of CF_3_COOH with the free phenol (Figure 20). Later, Zhou et al. [47] developed an aqueous C–H *ortho*-aminomethylation of free phenols **124** with trifluoroborates **125** by iodine catalysis, affording various functionalized phenol derivatives **126** in 20–98% yields. Based on controlled experiments, the authors proposed that the aminomethyl radical intermediate **127** was involved in the reaction pathway by oxidation of the hypoiodous acid (Figure 20).

## 3. C3-Functionalization

The *meta* functionalization of free phenols is challenging because the phenolic hydroxyl group can not only activate the aromatic ring for aromatic electrophilic substitution, but also guide its *ortho* or *para*-substitution [48,49,50]. Besides, the *meta*-sites of phenol are not easily activated by metal chelation [51,52]. As a result, the *meta*-C–H functionalization of free phenols could not be achieved via existing methods based on conventional C–H functionalization, electrophilic aromatic substitution, or oxidative coupling. Recently, Senior et al. [53] reported a *meta*-selective C–H arylation of polysubstituted free phenols **130** via regiodiversion of electrophilic aromatic substitution. This method achieved the *meta*-selective C–H arylation of sterically congested phenols through a Bi(V)-mediated electrophilic arylation and a subsequent aryl migration/rearomatization. A variety of the products of the *meta*-selective C–H arylation of phenols **135**, including the electron-donating and electron-withdrawing groups, were obtained in excellent yields. A preliminary mechanistic study suggested that this reaction involved a phenonium ion intermediate **133** (Figure 21).

## 4. C4-Functionalization

The electron-donating hydroxyl group and the reduced steric of C4 relative to either the C2 or the C6 of free phenols are capable of promoting the *para*-selective C–H functionalization of free phenols. Here, we introduced the *para*-selective C–H functionalization of free phenols according to metal catalysis, electrocatalysis, and other methods.

### 4.1. Metal-Catalyzed C–H Bond Functionalization of Free Phenol

In 2018, Bhanage and co-workers [54] reported a palladium-catalyzed aerobic oxidative carbonylation of a C–H bond of free phenols **136** for the synthesis of *p*-hydroxybenzoates **137** in 50–76% yields using molecular oxygen as a terminal oxidant. The substrates bearing the electron-withdrawing *meta*-nitro, *meta*-trifluoromethyl groups on phenol were not compatible with this reaction system and no corresponding products were observed. Besides, *p*-cresol with methanol also failed to react under the optimized reaction conditions. Based on controlled experiments, the authors proposed that the carbonylation reaction involved the in-situ production of an intermediate 4-iodophenol **138** that subsequently underwent the nucleophilic attack of alcohol to offered *p*-hydroxybenzoate **137** via reductive elimination of palladium metal (Figure 22).

Later, a gold-catalyzed highly chemo- and regioselective C–H bond functionalization of free phenols **142** with haloalkynes **143** was developed at room temperature by Hashmi’s group [55]. A series of the products of *para*-C–H functionalization instead of OH-additions **144** were obtained in good to excellent regio-selectivity and diastereoselectivity, which was attributed to the catalyst and the base-free condition in this reaction. When the preferred position was blocked, the *ortho*-C–H functionalization of free phenols was feasible. A preliminary mechanistic study suggested that the gold catalyst directly coordinated the chloroalkyne carbon, which formed the more stable π-activated alkyne **145**/vinylcation **146**. Then, the C-4 position of free phenols attacked the highly electrophilic alkyne instead of O-H insertion to generate β-haloalkenes **144** (Figure 23).

In 2021, the Liu group [56] reported a regiospecific and site-selective C–H allylation of free phenols **148** with vinyldiazo compounds **149** catalyzed by In(OTf)_3_. A series of phenol-allylation products instead of OH-additions **151** were obtained with 50–99% yields (Figure 24). The reactions of aryl substituted vinyldiazoacetates with phenols provided the *para*-C–H allylation products, but alkyl-substituted vinyldiazoacetates transformed the *ortho*-selective-C–H allylation products under the standard catalytic conditions. Based on controlled experiments, the authors proposed that this allylation reaction involved a carbene intermediate **150**, which was generated during the reaction process by In(OTf)_3_ releasing of N_2_.

### 4.2. Electrochemical C–H Bond Functionalization

Waldvogel’s group [57] reported an efficient electrochemical synthesis by dehydrogenative coupling of free 2,6- or 2,5-substituted phenols under metal catalyst- and oxidant-free conditions in 2019. A variety of the desired 4,4′-biphenols **154** were obtained with 10–77% yields by anodic dehydrogenative cross- and homo-coupling in an undivided beaker-type cell equipped with two boron-doped diamond (BDD) electrodes at a constant current of 5.7 mA/cm^2^, using Bu_3_NMeO_3_SOMe as the electrolyte in 1,1,1,3,3,3-hexafluoropropan-2-ol with 0–6 vol.% water (Figure 25). A preliminary mechanistic study suggested that this reaction involved a radical intermediate produced by the anodizing of the phenol at the anode, which was subsequently trapped by the phenolic component.

In 2022, Banerjee and co-workers [58] developed an efficient electrochemical approach to the Ritter-type reaction at the C(sp^2^)–H of free phenols **155** in the presence of acetonitrile **156** for the direct synthesis of paracetamol **157** under exogenous oxidant- and catalyst-free conditions. The amination reaction was conducted in an undivided cell equipped with a graphite anode and a nickel cathode at a constant current of 6.0 mA, using *n*-Bu_4_NPF_6_ as the electrolyte at room temperature (Figure 26). The substrate range of this reaction was mainly free phenol derivatives, and the naphthols (*α* and *β*) and benzyl cyanide were not compatible with these reaction conditions. Based on the results of this reaction, the electrochemical C–H amination reaction was proposed to initiate via anodic oxidation of the phenol to generate carbocation **158**, which underwent classical Ritter steps in the presence of acetonitrile **156** to deliver the desired product **157**.

Recently, the Banerjee group [59] further developed an electrochemical sulfinylation of free phenols **162** with sulfides **163** under catalyst- and oxidant-free conditions. The aromatic sulfoxides **164** were obtained with 38–65% yields in an undivided cell equipped with a graphite anode and a nickel cathode at a constant current of 15 mA, using *n*-Bu_4_NPF_6_ as the electrolyte at room temperature (Figure 27). The phenols bearing the electron-withdrawing groups such as nitro, ester, and aldehyde, were unsuccessfully employed as the reaction substrates under standard conditions. Control experiments suggested that the electrochemical sulfonated reaction was proposed to initiate via the anodic oxidation of phenol to generate carbocation, which was then attacked by the nucleophilic sulfide to generate the desired product.

### 4.3. Others

In 2020, Procter and co-workers [60] revealed an efficient transition-metal free C–H/C–H-type cross-coupling of free phenols **166** with benzothiophenes **165** to form C2/C3 arylated benzothiophenes **167** with 34–95% yields in trifluoroacetic acid using TFAA as an activator. A variety of substrates, including C2-substituted benzothiophene *S*-oxides and C3-substituted benzothiophene *S*-oxides, were successfully used in the arylated reaction to realize the C–H bond functionalization of free phenols. However, when using 4-methoxyphenol as the substrate, the *ortho*-difunctionalization of 4-methoxyphenol was observed to give the corresponding product in 85% yield. Based on controlled experiments, the authors proposed that this reaction was initiated by reaction of the benzothiophene *S*-oxides with TFAA to give the activated sulfoxides **168**, which could engage directly at the *para*-position of phenol to generate the corresponding products by a vinylogous Pummerer-type mechanism (Figure 28).

Later, a triflic acid-catalyzed chemo- and site-selective C–H bond functionalization of free phenols **170** with 1,3-dienes **169** was realized by Liu et al. [61] in 2020. A series of *para*-selective allyl phenols **171** were obtained with 40–88% yields under mild conditions in a site-selective manner. However, the *ortho*-selective allylic alkylation of phenols with 1,3-dienes were observed under standard conditions in this transformation, and the *ortho*-C–H bond allylic alkylation product was obtained with 40% yield when the *p*-cresol was employed as the substrate to react with diene. The authors proposed that this HOTf-catalyzed allylic alkylation involved an ion-pair **172** including allylic cation and triflate anion, which was pushed away by the phenolic hydroxyl, owing to its steric hindrance to giving the *para*-selective product (Figure 29).

## 5. *Ortho*-Functionalization-Cyclization Process

There were also some examples in which the -OH group took part in further transformations. Due to the 1C,3O-bisnucleophilic reactivity, free phenols have been applied to a variety of [3 + n] cycloaddition reactions with biselectrophilic compounds for the synthesis of various heterocyclic compounds [62,63,64]. 

In 2019, Tang et al. [65] developed a recyclable nickel-catalyzed C–H/O–H dual functionalization of mandelic acids **173** with free phenols **174**. Various of 3-aryl benzofuran-2((3*H*)-ones **175** were obtained with 31–97% yields under solvent-free conditions. The authors proposed a possible pathway for dual activation of phenols (Figure 30). Intermediate **176** was generated from mandelic acid in the presence of Ni(OTf)_2_. Then, an inter-molecular Friedel-Crafts alkylation of **176** with phenol took place to give intermediate **177**, which underwent intramolecular esterification to give the corresponding product **175**. This method can be enlarged and applied to synthesize antioxidant Irganox HP-136 in an excellent yield.

In 2020, Li et al. [66] developed a base promoted diastereoselective [3 + 3] cycloaddition reaction of 2-arylideneindan-1,3-diones **178** with free *β*-naphthols **179**. Various functionalized pentacyclic indeno[1,2-*b*]chromen-(4b*H*)-ones **180** were synthesized with 47–99% yields under mild conditions (Figure 31). Splenocyte cells were treated with the synthesized compounds, and cell viability was determined by CCK-8 assay. These tested compounds showed selective inhibitive activity against ConA-induced *T*-cell proliferation.

Subsequently, the Ren group [67] reported a silver(I)-mediated cascade reaction of 2-(1-alkynyl)-2-alken-1-ones **181** with free 2-naphthols **182**. A series of *ortho*-functionalization-cyclization 1,2-dihydronaphtho[2,1-*b*]furans **183** were obtained with 53–84% yields (Figure 32). The controlled experiments indicated that the silver trifluoroacetate first coordinated with the triple bond of 2-(phenylethynyl)cyclohex-2-en-1-one to give the incomplex **184**, which then formed furan intermediate **185**. The naphthalen-2-ol attacked the intermediate **185** to afford furan compound **186** and regenerated silver trifluoroacetate. In the presence of AgTFA and oxygen, the furan ring in compound **186** was further oxidized to 2-buten-1,4-dione intermediate **187**. Finally, the 1,2-dihydronaphtho[2,1-*b*]furan **183** was obtained through a highly regio- and diastereoselective oxa-Michael addition.

In addition, there were some reactions in which C–H functionalization outside the original phenol ring. For example, in 2019, Yao group [68] reported a palladium-catalyzed remote 1,*n*-arylamination of unactivated terminal alkenes with aryl iodides and arylamines. In 2020, Wu’s group [69] developed a quadruple C–H activation coupled to hydro functionalization and C–H silylation/borylation based on weakly coordinated palladium catalyst. The mechanisms of both reactions involved the coordination of phenol with palladium and activation of C–H bond outside the original phenol ring.

## 6. Conclusions

This review has described recent progress in regioselective C–H bond functionalization of free phenols, which are readily available chemical feedstocks in organic synthesis. We have identified a number of methodologies which have been made to overcome longstanding challenges of selectivity in the field of the C–H activation of free phenols without directing groups. These include the *ortho*-, *meta*-, and *para*- selective C–H functionalization of free phenols, which have been successfully achieved by transition-metal catalysis such as Cu, Co, Bi, Pd, Ag, Au and In, electrocatalysis, photocatalysis, and so on, leading to the productions of many useful functionalized phenols.

Although significant progress has been made, there are still some challenges to the regioselective C–H bond functionalization of free phenols, including as below: (1) there are few reaction types, especially the *meta*- selective C–H functionalization of free phenols, which remains an important area for future development; (2) how to make proper use of electrocatalysis or photocatalysis to avoid the use of a lot of oxidants and chemical reagents, which will be more efficient and environmentally friendly in organic synthesis; (3) the asymmetric C–H bond functionalization of free phenols is rarely developed, which is a very interesting area of research. With regard to these challenges, we hope that the explanation of the regioselective C–H bond functionalization of free phenols in this review will provide useful guidance for further development methods for C–H activation of free phenols to proceed in efficient, simple, mild and environmentally friendly conditions.

## Data Availability

Not applicable.

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
