# Peer review of "Recent Advances in Regioselective C–H Bond Functionalization of Free Phenols"

_molecules, 2023, doi:10.3390/molecules28083397_

Round 1

Reviewer 1 Report

The phenolic compounds are typical substrate-directable molecules, this manuscript focuses on this topic with recent examples are of important significance to the field, and most of the discussed examples are latest. But I found the reaction types are not diverse, in which only focused on the C-H functionalization at original phenol ring, and none of the -OH groups take part in further transformations. This reviewer therefore consider the manuscript needs major revision to diversify the reactivity enabled by phenolic -OH group. Especially for widely reported ortho-functionalization-cyclization process (eg. Org. Lett. 2017, 19, 1318; Org. Lett. 2017, 19, 2917; J. Org. Chem. 2018, 83, 2173 etc.), and necessary discussions of C-H functionalization outside the original phenol ring (eg. Nat. Commun. 2020, 11, 5662; ACS Catal. 2019, 9, 4196).

Author Response

Reviewer: 1

Comments:

The phenolic compounds are typical substrate-directable molecules, this manuscript focuses on this topic with recent examples are of important significance to the field, and most of the discussed examples are latest. But I found the reaction types are not diverse, in which only focused on the C-H functionalization at original phenol ring, and none of the -OH groups take part in further transformations. This reviewer therefore consider the manuscript needs major revision to diversify the reactivity enabled by phenolic -OH group. Especially for widely reported ortho-functionalization-cyclization process (eg. Org. Lett. 2017, 19, 1318; Org. Lett. 2017, 19, 2917; J. Org. Chem. 2018, 83, 2173 etc.), and necessary discussions of C-H functionalization outside the original phenol ring (eg. Nat. Commun. 2020, 11, 5662; ACS Catal. 2019, 9, 4196).

Answer:

Thanks for the reviewer’s comments. In the revised manuscript, we summarized ortho-functionalization-cyclization of free phenols and discussed of C-H functionalization outside the original phenol ring.

Org. Lett. 2017, 19, 1318; Org. Lett. 2017, 19, 2917; J. Org. Chem. 2018, 83, 2173; Nat. Commun. 2020, 11, 5662; ACS Catal. 2019, 9, 4196" have been cited in the references.

Reviewer 2 Report

The manuscript, “Recent Advances in Regioselective C–H Bond Functionalization of Free Phenols" reports current knowledge and recent advances in ortho-, meta-, and para-selective C–H functionalization of free phenols, which are widely used in agrochemicals, pharmaceuticals, and functional materials. The manuscript is well-written, and numerous related methodologies have been elaborated.

Therefore, the manuscript in its present form could be considered for publication.

Author Response

Reviewer: 2

Comments:

The manuscript, “Recent Advances in Regioselective C–H Bond Functionalization of Free Phenols" reports current knowledge and recent advances in ortho-, meta-, and para-selective C–H functionalization of free phenols, which are widely used in agrochemicals, pharmaceuticals, and functional materials. The manuscript is well-written, and numerous related methodologies have been elaborated.

Therefore, the manuscript in its present form could be considered for publication.

Answer:

Thanks for the reviewer’s comments.

Reviewer 3 Report

This manuscript by Yanan Li et al. describes a review for recent advances in regioselective C-H bond functionalization of free phenols. Since the phenols are an important class of synthetic building blocks and compounds presented in a wide range of pharmaceuticals, bioactive molecules and functional materials. Many researchers focused their efforts on developing facile and mild synthetic methods for regioselective C-H bond functionalization of free phenols. Although during 2018-2022, there are about 5 reviews reported by different groups as shown as ref 11-15, most of reviews only focused on transition-metal-catalyzed functionalization of C-H bonds in phenols. Only one review reported by Zhai and co-workers discussed the recent advances in catalytic oxidative reactions of phenols and naphthalenols (ref 15). Therefore, there is still a demand for an update in this field especially discussed about the regioselective C-H bond functionalization of free phenols not just focused the transition-metal-catalyzed chemistry. The authors have also made some contributions the field of C-H bond functionalization of aromatic compounds, which makes it suitable for them to summarize the recent works in this similar field. Besides, this review provided a good introduction to this field and served as a springboard for further exploration. In general, this manuscript is well-organized and of broad general interest, this reviewer recommends the manuscript for publication in Molecules after some revisions.

1. A generally issue of the schemes, they are not unified. Such as the scheme 1 and scheme 3 are not reach the publishable quality picture, some details are too blurred to view (like the Cun+1 part). In the opposite, the scheme 4 is much clearer. Please make sure draw all the schemes and figures in one way and then paste as the same size in the paper.

2. Page 3, line 64, add the section number 2 before the section title. The section title should be “C2-Functionalization” instead of “C1- Functionalization”.

3. Page 3, line 81, use “loss of hydrogen atom via a single...” instead of “loss of hydrogen via a single…” would be better.

4. Please add the full name to all the acronym when it first shows in the paper. Such as page 4, line 88, use “Di-tert-butyl peroxide (DTBP)” instead of just “DTBP”. Please check the whole paper and add it.

5. Scientific langue is more about number instead of “moderate to excellent yields”, please at least list all the yields range numbers under the desired products in all the schemes. Just like what you did in scheme 4.

6. In the paper, all the reaction conditions should be draw in a consistent manner. Such as: “rt”, in scheme 5, “r.t.” in scheme 7; “equiv.” in scheme4, “equiv” in scheme 20.

7. Page 6, line 129, “Ji et al” should either be “Ji and co-workers” or “Zhao et al”. Due to in the describable way of “xx et al”, xx should be first author, and in “xx and co-works” or “xx group”, xx should be the correspond authors. Please check the whole paper.

8. Page 6, line 132-line 135, “Interestedly, the para-benzomorpholine 132 substituted phenol was also compatible with the reaction system to transform into the 133 desired C2-site selective aminated product in 58% yield with oxidation of the benzomor-134 pholine group to 4H-benzo[b][1,4]oxazine.” Please add the relate transform in the scheme 7.

9. Page 7, line 140-line 152, two paper were discussed. Few things need to be changed.

a. Line 141, “In 2018, Khakyzadeh and co-workers” should be “In 2018, Khakyzadeh, Zolfigol and co-workers”

b. Line 145, “In addition, 2-naphthol” is not correct, since both 1-naphthol and 2-naphthol were compatible.

c. Line 147, “Khakyzadeh and co-workers” should be “Rostami, Khakyzadeh, Zoligol, Taherpour and co-workers” or describe in a different way.

d. These two parts of work should be describe as formal C-H functionalization of free phenols due to the real C-H functionalization substrate is the protected phenol, and then deprotection give the free phenol C-H functionalization products.

e. The whole mechanism is totally wrong in both of the two initial works, please just delete the last sentence of this paragraph “And the 151 key radicals were reduced by O2 and K2S2O8, respectively.”

f. In the scheme 8, two conditions names “Zoligol’s condition” and “Khakyzadeh’s conditions” should be match the text part, or maybe use the publish years to distinguish them.

10. Page 9, scheme 10, left bottom, the benzothiophene oxidized by anode give 63, do not need to lose the proton.

11. Page 15, line 309, add section number 3 before the section title.

12. Page 16, line 325, add section number 4 before the section title.

13. Page 17, scheme 22 and page 18 scheme 23 the catalytic cycle need to draw more beautifully as a real cycle shape.

14. Page 23, line 540, “Liang, X.-A.” should be “Liang, X.”

Author Response

Reviewer: 3

Comments:

This manuscript by Yanan Li et al. describes a review for recent advances in regioselective C-H bond functionalization of free phenols. Since the phenols are an important class of synthetic building blocks and compounds presented in a wide range of pharmaceuticals, bioactive molecules and functional materials. Many researchers focused their efforts on developing facile and mild synthetic methods for regioselective C-H bond functionalization of free phenols. Although during 2018-2022, there are about 5 reviews reported by different groups as shown as ref 11-15, most of reviews only focused on transition-metal-catalyzed functionalization of C-H bonds in phenols. Only one review reported by Zhai and co-workers discussed the recent advances in catalytic oxidative reactions of phenols and naphthalenols (ref 15). Therefore, there is still a demand for an update in this field especially discussed about the regioselective C-H bond functionalization of free phenols not just focused the transition-metal-catalyzed chemistry. The authors have also made some contributions the field of C-H bond functionalization of aromatic compounds, which makes it suitable for them to summarize the recent works in this similar field. Besides, this review provided a good introduction to this field and served as a springboard for further exploration. In general, this manuscript is well-organized and of broad general interest, this reviewer recommends the manuscript for publication in Molecules after some revisions.

  1. A generally issue of the schemes, they are not unified. Such as the scheme 1 and scheme 3 are not reach the publishable quality picture, some details are too blurred to view (like the Cun+1part). In the opposite, the scheme 4 is much clearer. Please make sure draw all the schemes and figures in one way and then paste as the same size in the paper.

Answer: Thank you for your nice reminder. All schemes and figures have been modified in one way and then pasted as the same size in the paper in the revised manuscript.

  1. Page 3, line 64, add the section number 2 before the section title. The section title should be “C2-Functionalization” instead of “C1- Functionalization”.

Answer: In Page 3, line 64, “C1- Functionalization” was changed to “C2-Functionalization” in the revised manuscript.

  1. Page 3, line 81, use “loss of hydrogen atom via a single...” instead of “loss of hydrogen via a single…” would be better.

Answer: In Page 3, line 81, “atom” was added in the revised manuscript.

  1. Please add the full name to all the acronym when it first shows in the paper. Such as page 4, line 88, use “Di-tert-butyl peroxide (DTBP)” instead of just “DTBP”. Please check the whole paper and add it.

Answer: The full name has been added to all the acronym when it first shows in the paper. And the whole paper was checked and revised.

  1. Scientific langue is more about number instead of “moderate to excellent yields”, please at least list all the yields range numbers under the desired products in all the schemes. Just like what you did in scheme 4.

Answer: All the yields range numbers under the desired products were added in all the schemes.

  1. In the paper, all the reaction conditions should be draw in a consistent manner. Such as: “rt”, in scheme 5, “r.t.” in scheme 7; “equiv.” in scheme4, “equiv” in scheme 20.

Answer: Thank you for your nice reminder. All the reaction conditions have been drawn in a consistent manner.

  1. Page 6, line 129, “Ji et al” should either be “Ji and co-workers” or “Zhao et al”. Due to in the describable way of “xx et al”, xx should be first author, and in “xx and co-works” or “xx group”, xx should be the correspond authors. Please check the whole paper.

Answer: Thank you for your nice reminder. The whole paper was checked and revised.

  1. Page 6, line 132-line 135, “Interestedly, the para-benzomorpholine 132 substituted phenol was also compatible with the reaction system to transform into the 133 desired C2-site selective aminated product in 58% yield with oxidation of the benzomor-134 pholine group to 4H-benzo[b][1,4]oxazine.” Please add the relate transform in the scheme 7.

Answer: The relate transform was added in the scheme 7.

  1. Page 7, line 140-line 152, two paper were discussed. Few things need to be changed.
  2. Line 141, “In 2018, Khakyzadeh and co-workers” should be “In 2018, Khakyzadeh, Zolfigol and co-workers”

Answer: In page 7, line 141, “In 2018, Khakyzadeh and co-workers” was changed to “In 2018, Khakyzadeh, Zolfigol and co-workers.” in the revised manuscript.

  1. Line 145, “In addition, 2-naphthol” is not correct, since both 1-naphthol and 2-naphthol were compatible.

Answer: In page 7, line 145, “2-naphthol was” was changed to “2-naphthol and 1-naphthol were” in the revised manuscript.

  1. Line 147, “Khakyzadeh and co-workers” should be “Rostami, Khakyzadeh, Zoligol, Taherpour and co-workers” or describe in a different way.

Answer: In page 7, line 147, “Khakyzadeh and co-workers” was changed to “Khaef et al.” in the revised manuscript.

  1. These two parts of work should be describe as formal C-H functionalization of free phenols due to the real C-H functionalization substrate is the protected phenol, and then deprotection give the free phenol C-H functionalization products.

Answer: Thank you for your nice reminder. These two parts of work has been described as formal C-H functionalization of free phenols in the revised manuscript.

  1. The whole mechanism is totally wrong in both of the two initial works, please just delete the last sentence of this paragraph “And the 151 key radicals were reduced by O2and K2S2O8, respectively.”

Answer: Thank you for your nice reminder. “And the 151 key radicals were reduced by O2 and K2S2O8, respectively.” was deleted in the revised manuscript.

  1. In the scheme 8, two conditions names “Zoligol’s condition” and “Khakyzadeh’s conditions” should be match the text part, or maybe use the publish years to distinguish them.

Answer: Thank you for your nice reminder. The publish years were used to distinguish them in the scheme 8.

  1. Page 9, scheme 10, left bottom, the benzothiophene oxidized by anode give 63, do not need to lose the proton.

Answer: The scheme 10 was revised in the revised manuscript.

  1. Page 15, line 309, add section number 3 before the section title.

Answer: In page 15, line 309, the “3” was added before the section title in the revised manuscript.

  1. Page 16, line 325, add section number 4 before the section title.

Answer: In page 16, line 325, the “4” was added before the section title in the revised manuscript.

  1. Page 17, scheme 22 and page 18 scheme 23 the catalytic cycle need to draw more beautifully as a real cycle shape.

Answer: In Page 17 scheme 22 and page 18 scheme 23, the catalytic cycle was revised in the revised manuscript. The whole schemes were checked.

  1. Page 23, line 540, “Liang, X.-A.” should be “Liang, X.”

Answer: In page 23, line 540, “Liang, X.-A.” was changed to “Liang, X.” in the revised manuscript.

Round 2

Reviewer 1 Report

The manuscript has been improved according to the comments, this referee believe the current version is suitable for publication.